# Autonomic nervous system responses of dogs to human-dog interaction videos

**Shohei Matsushita[1], Miho Nagasawa[1,2], Takefumi Kikusui[1,2]***

**1** Department of Veterinary Science, Azabu University, Sagamihara, Kanagawa, Japan, **2** Center for Human and Animal Symbiosis Science, Azabu University, Sagamihara, Kanagawa, Japan

* takkiku@carazabu.com

**Data Availability Statement:** All relevant data are within the paper and its Supporting Information files.

**Funding:** This study was financially supported by the Grant-in-Aid for Scientific Research on

## Abstract

We examined whether dogs show emotional response to social stimuli played on videos. Secondary, we hypothesized that if dogs recognize themselves in videos, they will show a different emotional response to videos of self and other dogs. We compared heart rate variability among four video stimuli: a video of the owner ignoring another dog (OW-A-IGN), a video of a non-owner interacting with another dog (NOW-A-INT), a video of the owner interacting with another dog (OW-A-INT), and a video of the owner interacting with the dog subject (OW-S-INT). The results showed that root mean square of the difference between adjacent R-R Intervals (RMSSD) and standard deviation of the R-R Interval (SDNN) were lower in NOW-A-INT and OW-S-INT than in OW-A-IGN. There was no statistical difference in the responses to OW-S-INT and OW-A-INT, suggesting that dogs did not distinguish themselves and other dogs in videos. On the other hand, the difference in mean R-R Interval between OW-S-INT and OW-A-INT showed positive correlation with the score of attachment or attention-seeking behavior. Therefore, this study does not completely rule out self-recognition in dogs and there remains the possibility that the more attached a dog to its owner, the more distinct the dog's emotional response to the difference between the self-video stimulus and the video stimulus of another dog. Further studies are needed to clarify this possibility.

## Introduction

Self-recognition is observed in humans between the ages of two to three years [1], and the development of this ability also correlates with the onset of high levels of empathy and altruistic behavior in humans [2]. Therefore, self-recognition is considered to be the basis of advanced and complex social formation in humans. The mark test has been commonly used as a method to determine whether an animal has the ability to recognize itself [3]. This method involves marking parts of the body of the target animal that the animal cannot see except in the mirror, and examining whether the animal can recognize the mark in the mirror. If the animal sees the mark in the mirror and attempts to frequently touch or examine the mark on its own body, then the animal is considered to have the ability of mirror self-recognition (MSR). Because the presence of MSR has long been confirmed only in humans and some non-human primates (ex. [3–5]), the evolutionary development of MSR was thought to be limited

Innovative Areas (No. 15K21739), Grant-in-Aid for Scientific Research (A) (No. 19H00972), Grant-in-Aid for Challenging Research (Exploratory) (No. 19K22823) from the Japan Society for the Promotion of Science, and JST, CREST Grant Number JPMJCR22P2, Japan. The funders had no role in study design, data collection and analysis, decision to publish, or preparation of the manuscript.

**Competing interests:** The authors have declared that no competing interests exist.

to lineages such as humans and other non-human primates. Recently, however, its presence has been confirmed in other animal species, such as magpies, dolphins, elephants, and fish [6–9], and it has begun to be proposed that self-awareness may be an example of convergent evolution, as it occurs independently in different species. In order to explore the detailed mechanisms by which this self-awareness has followed an evolutionary path and has emerged, a comparative study in a larger number of animal species is necessary.

Dogs are the oldest domesticated species and have social cognitive abilities that are similar to those of humans, such as understanding of human pointing, use of gaze, and acquisition of perspectives from others [10–12]. Additionally, dogs have shown empathy-like responses to humans or conspecifics [13–17]. However, no successful cases of the mark test have been reported [18]. Dogs may fail the mark test because of lack of motivation to investigate objects on their bodies. Horowitz [19] designed the "olfactory mirror" test as an alternative method of mark test, in which dogs were considered to be aware of their own odours. This is a remarkable result showing the possibility of self-recognition in the dog's olfaction. As Gallup and Anderson [20] pointed, however, if the species that primarily use visual ability for communication cannot achieve the MSR test, the species would not have self-recognition ability by other sensory functions. Dogs use visual ability and it is worth trying to conduct the MSR test. It will be important to construct new experimental methods to explore the distinction between self and others in the visual and other senses, instead of the mirror test, when examining these complex cognitive abilities in various animal species.

Therefore, in this study, we designed an experiment using videos as a new paradigm to explore self-cognitive abilities, as an alternative to the mark test. In humans, the response to the self of past video clips has been investigated. Lewis and Brooks-Gunn [21] compared children's reactions to own past videos, live videos, and videos of other children. The results revealed that the visual contingency appeared at around nine months of age, and that it became a cue to distinguish between the images of the self and others. By the age of two, recognition of the self is developed by the morphological features of the face, even if there are no real-time accompanying cues. In addition, Povinelli et al. [22, 23] modified the traditional mirror test and established a self-recognition task using videos of the self.

Although it is hard for dog to pay attention to images/videos, it has been suggested that dogs can distinguish between different emotional states and social information by the visual information from photographic stimuli (e.g. [24, 25]). If the dogs can recognize the social information from the videos, there should be different emotional responses elicited by different social contexts presented by the videos. However, the social response of dogs to videos has not been well investigated (e.g. [26, 27]). Therefore, the first aim of this study was to investigate changes in the autonomic nervous system associated with interactions that may induce emotion in dogs, and to ascertain whether dogs produce emotional changes from viewing videos in response to different social interactions. We focused on the jealousy paradigm, which we expect to drive autonomic nervous system activity in dogs [28]. In this context, the dogs see a situation in which their owner is interacting with another dog. Several previous studies have reported that dogs experience emotions such as jealousy in these contexts (e.g. [29, 30]). Therefore, we hypothesized that negative emotions would be elicited in dogs when they see a social interaction video between their owners and other dogs. In contrast, we predicted that dogs' emotions would not be induced if they see a non-social interaction video in which the owner ignores other dogs. The second aim is to clarify whether dogs can distinguish themselves from other dogs in videos. Although this ability cannot be described as a self-recognition that supports higher-order cognitive mechanisms related to metacognition and metamemory, it is worth examining in order to clarify the budding function of dogs' self-recognition. In this study, in addition to the video of the interaction between the owner and other dogs, the dogs

were shown a video of themselves interacting with their owners in the past. If they are familiar with their self-image through a mirror, it would also lead to habituation of the self-image of the video. One question that arises is whether the responses to the video of itself would differ based on whether dogs see their image in the mirror as itself or another subject. That is, if the dog becomes habituated with the self-image in the mirror and perceives the mirror image as another dog, the owner's interaction with the "familiar self in the mirror" in the video, such as receiving pets from the owner, will be viewed as the "other dog" interacting with the owner. It has been suggested that dogs show aversion (or jealousy-like behavior) when people interact with a familiar dog living in the same household, such as actively rewarding a familiar dog ([30–33] however, also note [34]). Therefore, if the dog recognizes the "familiar self in the mirror" as the "other dog," there will be no difference in the reactions to the videos of the subject's own interactions and those of other dogs, and aversion will be induced in both situations. If the "familiar self-image in the mirror" is different from the "other dog," then it is likely that there will be a difference in response between the "other dog" interaction with the owner and the "past video image of the self". As a control, the dog will not show such an emotional response when watching a "non-owners" interacting with the "other dog".

A recent study [35] has developed a method for assessing emotional changes in dogs using a physiological/neurological technique called heart rate variability (HRV) analysis. Several indicators are used as parameters of HRV. The root mean square of the difference between adjacent R-R interval (RRI) reflects the variability between beats of heart rate and is the primary time-domain measure used to estimate the vagal-mediated changes reflected in HRV (i.e., RMSSD). In contrast, both the sympathetic and parasympathetic systems contribute to the standard deviation of the RRI (i.e., SDNN). When the RRI is a series of the same values, the SDNN becomes small, and when it contains many different values, the SDNN becomes large. Katayama et al. [35] reported that dogs show a decrease in SDNN during positive stimuli and a decrease in RMSSD during negative stimuli. That is, the parameters of HRV are useful indicators in measuring the activity of the autonomic nervous system caused by emotional states [36, 37], and we predicted that when dogs see their owners interacting with other dogs, it will lead to a decrease in RMSSD.

In addition, this study used the Canine Behavioral Assessment and Research Questionnaire (C-BARQ), a questionnaire system that is considered to be an objective measure of a dog's temperament. Emotional changes in the jealousy are influenced by the temperament of the dog [28, 38, 39]. The attention-seeking behaviors in this C-BARQ trait include pushiness and "jealousy" when attention is given to third parties. Therefore, it is highly likely that a dog's social responses will be varied by its individual temperament. In this study, we predicted individual differences in emotional responses, such as jealousy and disgust, to social interaction videos by dogs, and investigated the relationship between these responses and dogs' attachment to their owners using C-BARQ. In the same way, we examined whether the individual difference of the changes in HRV in dogs were correlated to their aggressive or fear temperament. In particular, we predicted that when the HRV value of the condition in which the owner interacts with the self minus the condition in which the owner interacts with other individuals was used as the dependent variable, this difference would be larger for individuals with higher levels of attachment. In other words, this means that HRV is likely to change as a result of distinguishing between the self and other individuals and perceiving videos of other individuals as negative emotions.

## Methods

### Subjects

A total of 15 dogs with no known hearing or sight problems participated in this study. One dog was excluded because it was unable to participate in the entire two-day experiment, and

Table 1. Breed, sex, and age of the dogs tested in this study.

| Subject | | | | Other | | |
|---|---|---|---|---|---|---|
| ID | Breed | Sex | Age (years) | Breed | Sex | Age |
| A | Standard poodle | F | 10 | Saluki | F | 2 |
| B | Standard poodle | F | 3 | Saluki | F | 2 |
| C | Standard poodle | M | 3 | Saluki | F | 2 |
| D | Standard poodle | M | 3 | Saluki | F | 2 |
| E | Mix | M | 3 | Standard poodle | F | 3 |
| F | Bolognese | F | 5 | Standard poodle | F | 3 |
| G | Saluki | F | 2 | Standard poodle | F | 3 |
| H | Mix | F | 1 | Toy poodle | F | 12 |
| I | Mix (Chihuahua × Papillon) | M | 5 | Cavalier king charles spaniel | M | 8 |
| J | Cavalier king charles spaniel | M | 8 | Mix (Chihuahua × Papillon) | M | 5 |
| K | Yorkshire terrier | M | 3 | Mix | M | 3 |
| L | Toy poodle | M | 7 | Standard poodle | M | 3 |

two dogs were excluded because they could not accurately measure the ECG. As a result, dogs, comprising five females and seven males, with a mean age of 4.42±2.68 years old were included in this study (Table 1). All dogs were living as pets with their owners. Additionally, the owners confirmed that the dogs had seen themselves in the mirror in the past. We asked the owners to get the dogs used to looking in the mirror at their owners' homes and other places regularly for about two weeks before the experiment began. The dogs that participated in the experiment engaged calmly and non-aggressively with non-owners in general. All methods were carried out in accordance with relevant guidelines and regulations. Regarding animal experiments, the experimental procedures were approved by the Animal Ethics Committee of Azabu University (#180410–1) and regarding human experiments, the experimental procedures were approved by the Ethical Committee for Medical and Health Research Involving Human Subjects of Azabu University (#052). The consent of the owners, non-owners and experimenter was obtained after explaining the experiment procedures, and they were informed that they could withdraw at any time during the study. Informed consent was obtained to publish the information/image(s) in an online open-access publication from all the participants, and we confirmed that informed consent was also obtained from the experimenters/non-owners. In addition, all the figures with subjects in them are used with permission.

## Experiment environment and apparatus

The experiment was conducted in a room at Azabu University (Fig 1). A projector (PT-VW355N, Panasonic, JPN) was installed on the ceiling so that images could be projected on the screen. A speaker (SRS-X1, SONY, JPN) was placed near the screen. A fence (left and right: 155 cm x 110 cm, front: 100 cm x 180 cm) was installed around the screen to prevent the dogs from touching the screen. A video camera (HDR-CX680, SONY, JPN) was placed near the screen to record the experiment. Similarly, a video camera (HDR-CX680, SONY, JPN) was installed at the rear. The video was played by the experimenter from a PC (13Z970-ER33J, LG Electronics, South Korea) connected to the projector.

## Experimental conditions (event conditions)

To make the dogs experience more negative emotions, experimental conditions were designed with reference to and modification of the experimental contexts in previous studies [29, 34,

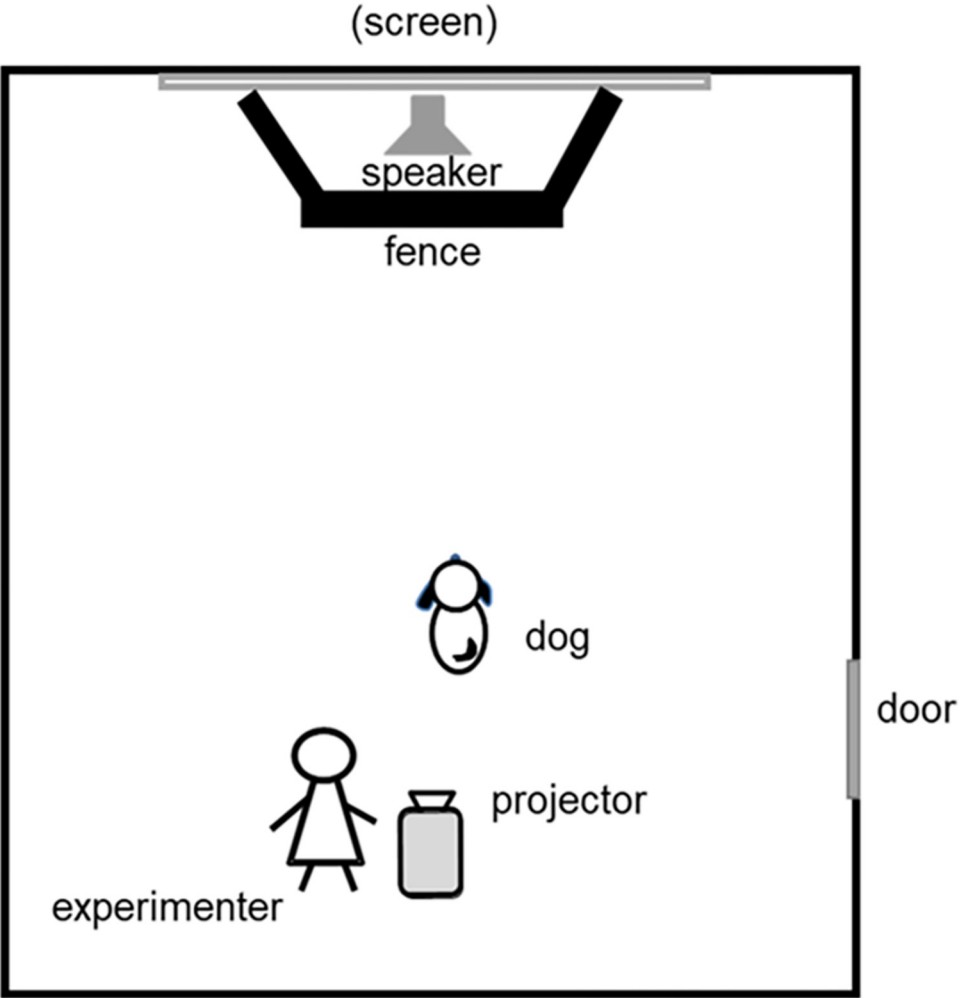

**Fig 1. An experiment environment.**

40]. Two types of video stimuli were prepared as an experimental group. One was a video (OW-A-INT) of the owner interacting with another dog. The other was a video (OW-S-INT) of the owner interacting with their own dog (subject). By comparing the responses to these video stimuli, we investigated whether dogs could distinguish themselves from another dog. We expected that if the dogs could distinguish between themselves and others, their responses to the two video stimuli would be different. In addition to these comparisons of video stimuli, two types of video stimuli were prepared as controls. One was a video (OW-A-IGN) of the owner ignoring another dog without interacting with it. By comparing these video stimuli with those of the experimental group, we investigated whether the dogs' responses were affected by differences in social interaction or by the owner's attention. The other was a video (NOW-A-INT) of a non-owner interacting with another dog. We investigated whether the dogs' responses, compared to the experimental group, could be influenced by the context of the human interaction, regardless of the specific person or the presence of the dog. A total of four types of video stimuli were used as event conditions in the experiment, and the sequence of the presentation of four video was pseudorandom.

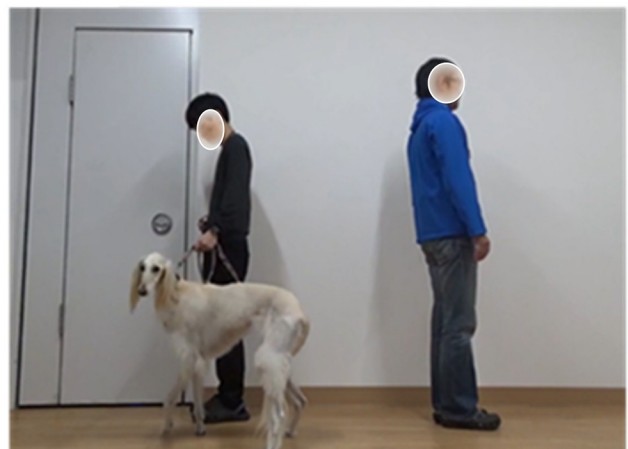 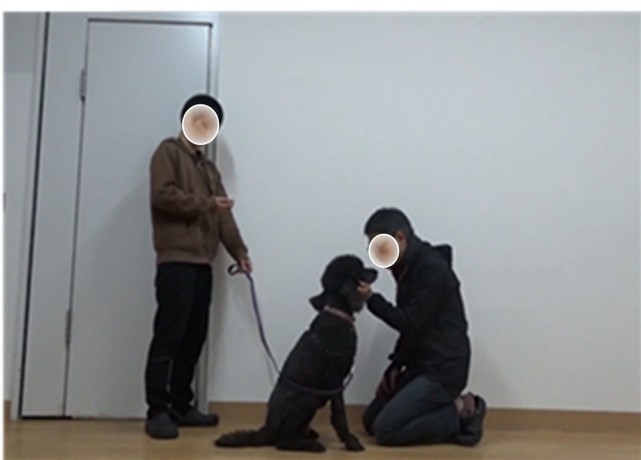

**Fig 2. A.** A scene from the OW-A-IGN. Enter and exit through the door on the left. The person on the left is the experimenter and retained the dog. The person on the right is the owner of the subject. In this case, the dog is not the owner's dog. **B.** A scene from the OW-A-INT, NOW-A-INT, OW-S-IN. Enter and exit through the door on the left. The person on the left is the experimenter. The person on the right is the owner or non-owner. In this case, the dog was either the dog subject or another dog.

## Video production

The videos used in the experiments were recorded using a video camera (HDR-CX680, SONY, JPN). Videos were recorded before the experiment, but only the owner-self videos were taken between the day of the experiment and about two weeks previously for the owners' convenience. If the video was taken on the day of the experiment, the experiment began approximately one hour later. When recording the video, it was adjusted so that it was projected onto the screen at actual-size during the experiment. During the video of the owner ignoring another dog, the owner did not speak at all and turned away from the dog (OW-A-IGN; Fig 2A). In contrast, when the person (owner/non-owner) interacted with the dog (self/other dog), the person was asked to interact with the other dog as if they were playing with their own dog (OW-A-INT, OW-S-INT, NOW-A-INT; Fig 2B). To minimize the possibility of unfamiliarity of the other dog in the comparison between familiar-self-video and familiar-other video, the other dog was a familiar one. A familiar one was a dog that has spent time sharing the same environment (University or home), showing affiliative behavior such as greeting to the person. We considered that a stranger dog would induce aggression and fear, so we used a familiar dog to elicit jealousy by using a familiar dog. For each subject, other dog was the same individual. Therefore, the breed and size of another dog could not be controlled. Non-owners were of the same gender as the dog owners and applied to those with whom they were familiar. This is because we considered that the application of strangers would result in HRV effects caused by interest in a stranger or aggression toward a stranger. These recorded videos were adjusted to five minutes in length for use in the experiment. Videos were recorded from when a person (owner/non-owner) entered the room for filming to when they left the room. The dog was accompanied by the experimenter until entering and exiting the video recording room. In the sequence of events, the experimenter was quiet except to signal the beginning and the end. In summary, the experimenter and the person (owner/non-owner) entered the video recording room simultaneously, the person (owner/non-owner) either interacted with or ignored the dog for a period of time, and then the experimenter and the person (owner/

non-owner) left the video recording room simultaneously. Therefore, during the experiment, the video projector was always turned on, and when the video stimulus was presented, the video was projected, and at other times, a still image of the empty video recording room was projected onto the screen.

### Heart rate measurement

The heart rate measurements in this study were performed based on a previous study [35]. Self-attaching bandages (1410 1404, 3M, USA) and disposable electrodes (G236, NIHON KOHDEN, JPN) and sponges (6 cm—5 cm—2.5 cm) were combined as shown in S1 Fig to allow the electrodes to be placed on the body surface of the dog without shaving. A sponge was sandwiched between them firmly to fix the electrodes so that they would not float, and noise would enter. An ELECTRODE GEL (15–69, Parker Laboratories, USA) was also added to the electrodes. Faros (emotion Faros 360˚, Bittium, Finland) was used as a device for recording the electrocardiogram of dogs and was finally attached as shown in S2 Fig. If the dog was unwilling to wear the electrocardiograph (ECG), the ECG was discontinued. All the dogs were not reluctant to wear the ECG.

### Procedure

First, the owner and the dog freely explored the room for five minutes to become habituated to the experimental environment and the experimenter. The dog was fitted with an electrocardiograph, and an additional five-minute acclimatization period was provided to avoid the effect of this wearing on the heart rate. After this, the owner left the room, leaving only the experimenter and the dog in the room. Then, a rest period of five minutes was provided with the experimenter to measure the resting heart rate (rest condition), and then the first video stimulus was presented for five minutes (event condition). The experimenter controlled the projector during the experiment. To ensure that the dog was aware of the video, experimenter played the stimulus when the dog was paying attention to the screen. The dogs were not restrained during the experiment in order to avoid the stress caused by restraint. Multiple cameras checked the dog's movements, mainly to see if the dog's head was facing the screen. After presenting the video stimulus, a five-minute rest period was provided to eliminate the effects of viewing the previous video. After that, a rest period of five minutes was taken with the experimenter again to measure the resting heart rate (rest condition). Following this, the second video stimulus was presented for five minutes (event condition). On a separate day (about a week later), using a similar procedure, two video stimuli that were different from the previous one were presented to the dog. Thus, we presented a total of four video stimuli in random order for each dog.

### C-BARQ assessment

We evaluated the usual behavior of dogs toward their owners. This is because, as mentioned previously, we are investigating the possibility that a dog's temperament affects its response to social interaction. Therefore, C-BARQ was used to evaluate the dog's behavior toward their owners [39]. C-BARQ data for this study were collected from dog owners before or after the experiment using the Japanese version of C-BARQ [41]. This involves asking owners to indicate how their dog has recently responded to various common events and stimuli using a scale of 0 to 4. The various items and subscale scores of the C-BARQ have been shown to provide an accurate measure of the dog's behavioral phenotype. The C-BARQ assessment yields multiple temperament factors in dogs. The study used scores for dog-directed aggression, dog-directed fear, and attachment or attention-seeking behavior. The reasons for using these indices are

that we are interested in the possibility that the subject dog is reacting to other dogs with strong interest (aggression or fear), not just that the strength of the attachment may incite more jealousy.

## HRV analysis

The R-wave was detected, and the RRI was calculated using MATLAB (www.mathworks.com) code as previously described [17], while visually checking the electrocardiogram waveform at the extracted experimental time. If there was noise in the ECG waveform that made it difficult to extract the R-wave, it was treated as a missing value (not-a-number). For the HRV analysis, the analysis time-bin was set as 15 continuous seconds. We were then able to obtain 20 time bins from about 5 minutes of data. HRV time-domain analysis was performed from these RRI. The mean RRI; the SDNN, which is an indicator of overall autonomic nervous system activity; and the RMSSD, which is an indicator of parasympathetic nerve activity, were calculated from the RRIs for each condition as HRV indices. In order to exclude the effects of body posture and movement on HRV, we extracted the areas where the same posture was maintained for more than 15 s between the conditions to be compared based on the labeling of successive behaviors and used them as HRV indices for analysis. The video data and HRV data were synchronized by video recording the sound emitted during HRV data recording and its internal clock. If there were multiple time bins that corresponded to each condition, the average of the corresponding time bins was calculated.

## Statistical analysis

Data analysis was performed using statistical analysis software R version 3.6.1 (https://www.r-project.org/). For each of the HRV indicator data, logarithmic transformation was performed to ensure normality. A Kolmogorov-Smirnov test indicated that the data are consistent with a normal distribution (RRI: $p = 0.53$; RMSSD: $p = 0.89$; SDNN: $p = 0.60$). Linear mixed-effects models (LMMs; R package "lmerTest" [42]) were used to compare the HRV values between rest conditions and between video stimuli. Conditions (rest conditions or event conditions) and Day (1day, 2day) were included as fixed effects. Subject's ID and age were included as random effects. If it was confirmed that there was no significant difference between the resting conditions, the mean of the four resting conditions per individual dog was calculated as the baseline value of the resting condition. The difference between the values of each video stimulus and the baseline values was then calculated and compared between the video stimuli using LMMs. Significant effects were further analyzed using post-hoc comparisons. A paired t-test with Holm method was used for the multiple comparison in post-hoc test. In order to confirm the association between the C-BARQ temperament scores and the HRV data, a multivariate analysis was conducted using the NOW-A-INT, OW-A-INT, OW-S-INT, and the difference between the OW-S-INT and OW-A-INT values of HRV as dependent variables and the C-BARQ scores (dog-directed aggression, dog-directed fear, and attachment or attention-seeking behavior) were used as explanatory variable. For multivariate analysis, linear regression analysis was used. In order to remove the effect of multicollinearity from causing major problems in the results when performing linear regression analysis, the correlation between the explanatory variables was checked and the variance inflation factor was calculated and confirmed to be less than 10.

## Results

LMMs (two factors: day and rest condition) were performed to check whether all four resting states before each video stimulus presentation were equal. The results showed that there was

no significant effect of day and rest conditions on all HRV indicators (see S1 Table). Therefore, the mean of the HRV values during each of the four rest conditions was calculated, and this value was taken as the baseline value during the rest condition. The difference between the HRV values and baseline values at each video stimulus was determined and compared between the video stimuli. Comparisons between video stimuli were then performed by the LMMs, and significant effects of different video stimuli were found for all HRV indicators (see S2 Table). Therefore, multiple comparisons between video stimuli were performed as post-hoc tests. The results showed no significant differences between video stimuli for meanRRI, but NOW-O-INT was significantly lower than OW-A-IGN for both RMSSD and SDNN (RMSSD: $df$ = 11, $t$ = -3.14, $p$ = 0.047, $r$ = 0.69, 95% Cl = -0.40 to -0.07; SDNN: $df$ = 11, $t$ = -3.18, $p$ = 0.044, $r$ = 0.71, 95% Cl = -0.31 to -0.06; Fig 3, see also S3 Table), and OW-S-INT was also significantly lower than OW-A-IGN for both RMSSD and SDNN (RMSSD: $df$ = 11, $t$ = -3.23, $p$ = 0.048, $r$ = 0.70, 95% Cl = -0.56 to -0.10; SDNN: $df$ = 11, $t$ = -3.34, $p$ = 0.040, $r$ = 0.71, 95% Cl = -0.38 to -0.08; Fig 3, see also S3 Table). There was no significant difference between the experimental groups, OW-A-INT and OW-S-INT, for all HRV indicators. We then analyzed whether the dog's temperament affected physiological responses to social interaction videos in the NOW-A-INT and OW-S-INT conditions, where there were significant differences in HRV data. For these two conditions (NOW-A-INT, OW-S-INT), linear regression analysis was performed with HRV data as the dependent variable and C-BARQ score as the explanatory variable. The analysis showed no significant differences in HRV and CARQ scores for both NOW-A-INT and OW-S-INT (Tables 2 and 3). The linear regression analysis was also performed with the OW-S-INT data minus the OW-A-INT data as the dependent variable and the C-BARQ score as the explanatory variable. As a result of regression analysis, no significant correlation was found in RMSSD and SDNN (see S4 Table). However, the mean RRI was

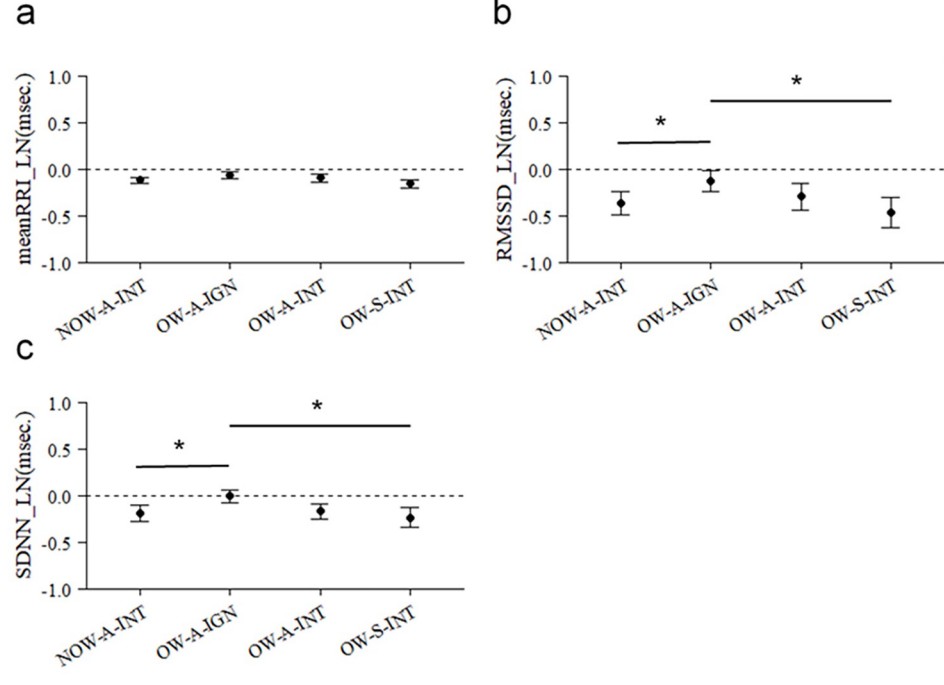

**Fig 3. Comparison between each video stimulus on the HRV index. a** mean RRI; **b** RMSSD; **c** SDNN are shown. The black circle represents the average value. Error bars represent standard errors (paired t-test: $^*p < 0.05$).

**Table 2. Parameter estimates for the relationship using HRV data as the dependent variable and CBARQ scores as the explanatory variable in the NOW-A-INT condition.**

| Condition: NOW-A-INT | | | | | |
|---|---|---|---|---|---|
| HRV | Explanatory variables | Estimate | Std. Error | t value | p value |
| meanRRI | dog-directed aggression | 0.63 | 0.26 | 2.14 | 0.065 |
| | dog-directed fear | -0.38 | 0.30 | -1.25 | 0.246 |
| | attachment or attention-seeking behavior | -0.27 | 0.29 | -0.94 | 0.374 |
| | R squared value | 0.20 | | | |
| RMSSD | dog-directed aggression | 0.39 | 0.35 | 1.14 | 0.289 |
| | dog-directed fear | -0.38 | 0.36 | -1.07 | 0.317 |
| | attachment or attention-seeking behavior | 0.05 | 0.34 | 0.14 | 0.890 |
| | R squared value | -0.12 | | | |
| SDNN | dog-directed aggression | 0.37 | 0.35 | 1.08 | 0.314 |
| | dog-directed fear | -0.38 | 0.36 | -1.06 | 0.319 |
| | attachment or attention-seeking behavior | 0.09 | 0.34 | 0.28 | 0.787 |
| | R squared value | -0.13 | | | |

significantly lower as the score of C-BARQ (attachment or attention-seeking behavior) increased ($R^2$ = 0.35, $p$ = 0.04; Fig 4; see also S4 Table).

## Discussion

In this study, we examined the response of the autonomic nervous system of dogs under four different visual conditions. The results showed that there were significant differences between the four stimuli. Specifically, RMSSD and SDNN were lower in NOW-A-INT and OW-S-INT than in OW-A-IGN (Fig 3). This is may be a result of higher arousal of dogs in the human-dog interacting stimuli. Previous studies [35–37] have shown that changes in HRV parameters are useful indicators in measuring autonomic nervous system activity elicited by emotional states. In fact, dogs can distinguish between non-social images and social interaction images [43]. In line with previous study, the present results suggest that dogs are more sensitive to the social interaction video than to the non-social interaction video. Therefore, these results captured the changes in the emotional status caused by different stimuli. Thus, assessing HRV is a useful

**Table 3. Parameter estimates for the relationship using HRV data as the dependent variable and CBARQ scores as the explanatory variable in the OW-S-INT condition.**

| Condition: OW-S-INT | | | | | |
|---|---|---|---|---|---|
| HRV | Explanatory variables | Estimate | Std. Error | t value | p value |
| meanRRI | dog-directed aggression | 0.55 | 0.30 | 1.85 | 0.101 |
| | dog-directed fear | -0.40 | 0.31 | -1.31 | 0.225 |
| | attachment or attention-seeking behavior | -0.32 | 0.29 | -1.10 | 0.303 |
| | R squared value | 0.18 | | | |
| RMSSD | dog-directed aggression | 0.37 | 0.34 | 1.10 | 0.303 |
| | dog-directed fear | -0.47 | 0.35 | -1.34 | 0.217 |
| | attachment or attention-seeking behavior | 0.002 | 0.33 | 0.01 | 0.995 |
| | R squared value | -0.07 | | | |
| SDNN | dog-directed aggression | 0.45 | 0.33 | 1.36 | 0.210 |
| | dog-directed fear | -0.45 | 0.34 | -1.33 | 0.220 |
| | attachment or attention-seeking behavior | 0.04 | 0.32 | 0.13 | 0.903 |
| | R squared value | -0.03 | | | |

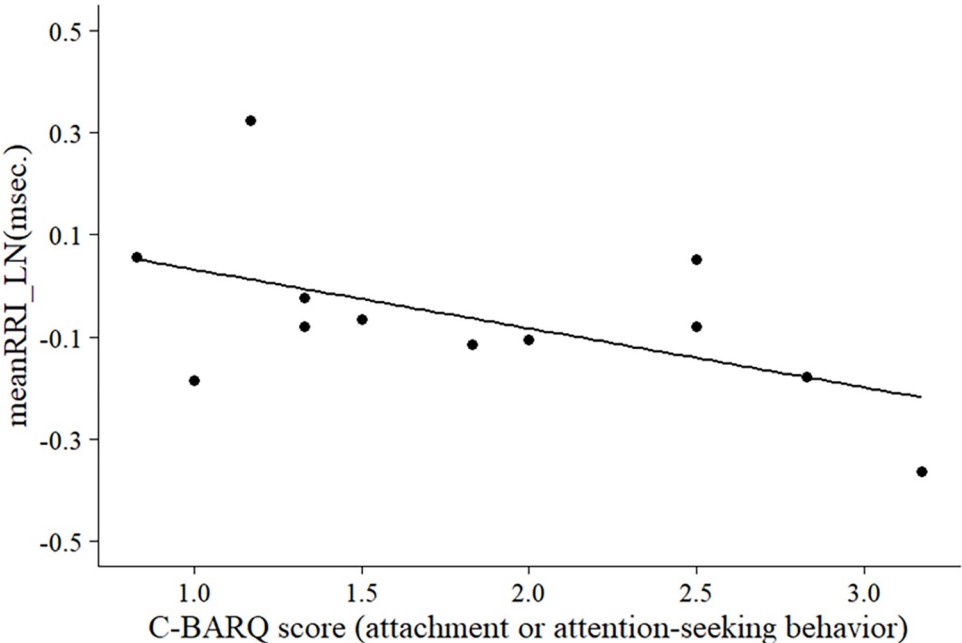

**Fig 4. Correlation between C-BARQ score (attachment or attention-seeking behavior) and mean RRI (linear regression; $\beta$ = -0.62, $p < 0.05$).**

method to detect changes of the dog's internal states in different social contexts. What do these emotional changes in dogs represent?

Previous studies have reported that increased heart rate, decreased RMSSD and decreased SDNN are associated with stress states [44, 45]. In this study, the effects of dog temperament were examined in each of the NOW-A-INT and OW-S-INT conditions, two conditions that were significantly different from the OW-A-IGN (control condition). The results showed that aggression and fear of dogs were not related to HRV in both the NOW-A-INT and OW-S-INT conditions. This suggests that it is unlikely that the dog's autonomic nervous system is activated due to the dog's potential aggression or fear toward other dogs. It should be noted, however, that the CBARQ may not have been assessed correctly by the owners and HRV parameters in the resting time could not exactly reflect neutral emotional state due to being left in an unfamiliar environment with the experimenter. Emotional changes elicited in this study are not jealousy-like emotions, but are more likely to be an expression of emotional changes originating from "fear" induced by the dog in the video. In particular, this may be due to the fact that in the NOW-A-INT and OW-S-INT conditions, the human was interacting with the dog and the dog in the video appeared to be more excited than normal to the dog subject. But in this case, it was difficult to explain why there was no difference between OW-A-INT and OW-A-IGN; OW-A-INT video also displayed human interaction with the dog. One way to solve this discrepancy was analyzing which part of the video the dog was paying attention to. Particularly in the 3 interaction conditions, dog were more attentive to the video stimuli as compared to the OW-A-IGN. However, because of the technical aspects of this study, we were unable to do so. Therefore, it is difficult to conclude that the dog's emotional change was caused by seeing the state of another dog, rather than the state of the human. In fact, it is possible that the presence of humans affected the HRV data in dogs, as studies have shown that dogs gazed longer at the human actors compared to dog social interaction images [43]. However, some studies suggest that dogs spontaneously prefer images of

conspecifics to human faces or inanimate objects [46], and that dogs are particularly attracted to representations of quadrupedal movement rather than humans [47]. Therefore, we suggest that the differences in HRV data between video stimuli were most likely influenced by the visual information obtained from the dog in the video.

As described above, the dog subjects showed vigilant behavior, such as paying attention to the video toward the dog in the OW-S-INT condition, suggesting that the dogs were unable to recognize the dog in the video as themselves. In fact, there was no significant difference in HRV data between the OW-A-INT and OW-S-INT conditions. The other dog used in this study was a dog familiar to the dog subjects. The dog in the mirror was also familiar to the dog subjects. Thus, the lack of difference in the autonomic nervous system response between these two stimuli may indicate that the dogs considered both stimuli to be equivalent. However, the possibility of dogs being able to recognize themselves cannot be completely ruled out. This is because, as shown in Fig 4, the meanRRI (OW-S-INT minus OW-A-INT) correlates with the score of C-BARQ (attachment or attention-seeking behavior). This suggests that the more attached a dog is to its owner, the more distinct the dog's emotional response to the difference between the self-video stimulus and the video stimulus of another dog. Under controlled experimental conditions, dogs [48, 49] and rats [50] showed increased heart rate, decreased parasympathetic activity, and decreased overall autonomic activity in a positive state of satisfaction with rewarding food or reunion with the owner after separation. Therefore, the fact that the meanRRI is shorter in dogs with higher attachment to their owners does not necessarily indicate a negative state, but may also reflect a higher emotional arousal including positive state for the dog.

Of course, the results of this study cannot determine whether dogs have self-recognition. This is because there are several critical concerns with this experimental setting. First, we were unable to examine the dog's reaction to the mirror. In this study, a two-week habituation period to the mirror was provided beforehand, but it was not clear whether the dogs were really allowed a sufficient period of time to realize that the image in the mirror was the self. In addition, we did not test whether the dog understood the nature of the mirror correctly. Therefore, it is necessary to quantitatively and time-dependent analyze the behavior of how dogs respond to the stimulus object reflected in the mirror during habituation. Second, it is still unclear whether dogs are capable of matching past images with their own. In the present study, we used dogs that were familiar with mirrors and had experienced a mirror image of themselves, to investigate whether they could distinguish between past images of the self and images of other individual dogs. This is because it is difficult to confirm whether dogs can visually recognize their own mirror images as in the mirror test in an experimental system. Although it is thought that human infants can recognize themselves from their own past images, we cannot yet accept the existence of this visual ability in dogs. In particular, studies of infants have shown that the detection of the relationship between the visual feedback of self-image and information about one's own movements is important in the developmental process of MSR [22, 51]. It is necessary to examine whether dogs can notice that their own movements are fed back to the video simultaneously, and the effect of delaying the video in such situations. Third, the present study used a context in which jealousy emotion was elicited, and it is possible that the lack of this emotion in the dog affected the results of this study. In other words, this context may not have motivated the dogs to elicit jealous emotion. And most importantly, the sample size was relatively small due to the pandemic. As S3 Table indicates, the effect size of the results of this study was not conclusive, but very suggestive, and needs to be validated with a larger sample size.

In conclusion, we found that dogs differ in their physiological responses to different video stimuli. In addition, the results of the present study did not provide clear results on the ability

of dogs to distinguish between self and others because of the small number of samples. However, we believe that this experimental design can effectively examine whether dogs respond in a way that distinguishes between themselves and others, in that it takes advantage of the characteristics of dogs and increases motivation to experiment. And by using physiological/neurological techniques, it will be possible to capture the slight differences in the dog's responses that are not apparent on the outside. This will hopefully lead to new research into the presence of MSR in dogs in the future. In particular, we were unable to clarify that dogs experience jealousy, as in some previous studies [34], which means that there is still a need for experiments that take into account dogs' temperaments. In the future, when examining such complex cognitive abilities in various animal species, it will be important to construct new experimental methods that explore the distinction between self and others in all senses, not just the mark test.

## Supporting information

**S1 Fig. Heart rate measurement procedure.** An electrode gel was added to the electrode. The negative electrode was placed slightly below the sternal peduncle. The grounding electrode is separated from the positive electrode. The positive electrode was attached around the processus xiphoideus.
(DOCX)

**S2 Fig. After the electrocardiograph is placed on the dog.**
(DOCX)

**S1 Table. Statistics of the results of the LMMs between resting conditions.**
(DOCX)

**S2 Table. Statistics of the results of the LMMs between event conditions.**
(DOCX)

**S3 Table. Statistics of multiple comparisons between event conditions by paired t-test.**
(DOCX)

**S4 Table. Parameter estimates for the relationship using HRV (OW-S-INT data minus the OW-A-INT data) data as the dependent variable and C-BARQ scores as the explanatory variable.** The explanatory variables are variables selected by the stepwise method.
(DOCX)

**S1 Data. Supporting data.**
(XLSX)

## Acknowledgments

We are deeply grateful to the laboratory members for their assistance in conducting this research, and to the owners and their dogs for their precious time to participate in this study.

## Author Contributions

**Conceptualization:** Shohei Matsushita.

**Data curation:** Shohei Matsushita.

**Formal analysis:** Shohei Matsushita.

**Funding acquisition:** Miho Nagasawa.

**Supervision:** Takefumi Kikusui.

**Validation:** Miho Nagasawa, Takefumi Kikusui.

**Visualization:** Shohei Matsushita.

**Writing – original draft:** Shohei Matsushita.

**Writing – review & editing:** Shohei Matsushita, Miho Nagasawa, Takefumi Kikusui.

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
