## [Decision Letter · Decision Letter 0]

21 Feb 2022

PONE-D-21-29226Autonomic Nervous System Responses of Dogs to Human-Dog Interaction VideosPLOS ONE

Dear Dr. Kikusui,

Thank you for submitting your manuscript to PLOS ONE. After careful consideration, we feel that it has merit but does not fully meet PLOS ONE’s publication criteria as it currently stands. Therefore, we invite you to submit a revised version of the manuscript that addresses the points raised during the review process.

As you see, all reviewers share the opinion, your study has merit. They appreciate that the method is original and highlights the relevance of using subtle physiological indicators of dogs’ emotions. On the other hand, they all are also critical to various parts of your manuscript.

Will you extend the literature also to quick examples of other species (see the comments of Reviewer 3)?

I agree with Reviewers 1 and 3 that the number of dogs used is a rather low. It’s a good idea to make a power analysis first, as suggested by Reviewer 1, otherwise the non-significant results were not convincing to such a few subjects.

There are many parts of the Methods to be clarified. In what order did you play the videos on the tested dog? (On random, in a pre-defined order?) You should design the statistical model for repeated measures, anyway.

It’s fine you mentioned the citation for the C-BARQ assessment. However, will you briefly describe the basic principles of this assessment for those who are not familiar with that method? Besides, I think that would be good for the other tests, too. For further details of the Methods improvement, follow the suggestions provided first by Reviewer 2. But many comments by the other two reviewers are also useful.

I have a problem myself with your statistics. You should first list all variables involved with some basic information, such as mean, SD, or SE, or min-max in countable variables and the levels of categorical variables. You haven’t mentioned potential effect of age, sex, and breed in your analysis (and how you coped with that). Before any other analyses, make the detection of multicollinearity. It’s most likely you did it so. However, when you mention multicollinearity as the last part of the Statistical analysis of the Methods, it is not clear how you did it. Still, I don’t think testing the multicollinearity by using the variance inflation factor only is sufficient. (See, for example, https://www.r-bloggers.com/2018/08/dealing-with-the-problem-of-multicollinearity-in-r/). For an extended period, statisticians have not recommended the stepwise method for studies like this (e.g., Derksen, S. & Keselman, H. J., 1992. Br. J. Math. Stat. Psychol. 45, 265-282; Whittingham, M. J., Stephens, P. A., Bradbury, R. B. & Freckleton, R. P., 2006. J. Anim. Ecol. 75, 1182-1189; etc.). Instead of using the stepwise procedure, you should first formulate the hypothesis (or hypotheses) according to which you should then construct the statistical model to be tested. (See for inspiration, for example Kleinbaum, D. G., Kupper, L. L., Muller, K. E. & Nizam, A., 2013. Applied regression analysis and other multivariable methods. Duxbury Press, Pacific Grove.)

I am afraid you will have to revise the manuscript substantially before I can judge it any further.

We look forward to receiving your revised manuscript.

Kind regards,

Ludek Bartos

Academic Editor

PLOS ONE

https://journals.plos.org/plosone/s/fileid=ba62/PLOSOne_formatting_sample_title_authors_affiliations.pdf".

“We have no competing interests.”

3.  We note that Figure [xxxx] includes an image of a [patient / participant / in the study].

Reviewers' comments:

Reviewer's Responses to Questions

**Comments to the Author**

1. Is the manuscript technically sound, and do the data support the conclusions?

Reviewer #1: Partly

Reviewer #2: Yes

Reviewer #3: Yes

2. Has the statistical analysis been performed appropriately and rigorously? 

Reviewer #1: Yes

Reviewer #2: Yes

Reviewer #3: Yes

3. Have the authors made all data underlying the findings in their manuscript fully available?

Reviewer #1: Yes

Reviewer #2: Yes

Reviewer #3: Yes

4. Is the manuscript presented in an intelligible fashion and written in standard English?

Reviewer #1: Yes

Reviewer #2: Yes

Reviewer #3: Yes

5. Review Comments to the Author

Reviewer #1: The manuscript is interesting and I liked the idea. I think it fits nicely on the topic of emotional response to social stimuli. Besides, it expands our knowledge on how dogs process and reacts to prerecorded videos.

There are a few things that are not quite clear to me but let's proceed with order.

INTRO:

The intro addresses the topic correctly. I believe that all the necessary literature has been cited and the initial hypothesis of the authors are clearly stated and easy to identify. Good idea to support behavioural data with a standardised questionnaire like the CBARQ.

METHODS:

Honestly, 12 is not a huge number and it would have been more interesting to have more subjects. I understand that sample size is an issue, especially during the pandemic but maybe I would have addressed this in the discussion. And/or you could add a power analysis showing that this sample is enough for the study.

Where the dogs intact or neutered? This factor should be also considered in the analysis.

Why did the authors not include also an unfamiliar dog? I would have definitely add more conditions (i.e., non-owner/owner interacting/non-interacting with unfamiliar dog).

Were non-owners friends of the owners, students, experimenters? It is not clear to me.

RESULTS:

It seems that here the results have been reported adequately.

DISCUSSION:

I have to admit that this section was quite hard to follow.

As far as I understand the authors' findings are the following:

No differences at all in the mean RRI,

RMSSD different between non-owner interacting with another dog and owner ignoring the other dog,

RMSSD different between owner ignoring the other dog and interacting with their own dog,

SDNN same as RMSSD

I would maybe make a table with such results in a way that it makes it easier for the reader to interpret your findings. And/or I would divide the initial section of the discussion in paragraphs.

Another thing that I think it is worth to mention in the discussion is that using questionnaires might be faulty to a degree. In previous studies employing the CBARQ the authors noticed how some people belonging to different cultures might have difficulties in expressing strong concepts like 'my dog NEVER/ALWAYS does X'. Hence, they tend to evaluate their pets in a more conservative way. I would add a line in the discussion addressing this.

Reviewer #2: The Autonomic Nervous System Responses of Dogs to Human-Dog Interaction Videos manuscript studies the emotional response to social stimuli played on videos as well as the ability of dogs to distinguish their own image from that of other dogs. Both topics are interesting and they need further investigation. The methodology is original and highlights the relevance of using subtle physiological indicators of dogs’ emotions. However, there are some concerns that I detailed below.

Abstract: please include the definition of RRI

Introduction

Please clarify and further explain this statement “no successful cases of the mark test have been reported”. You should describe previous studies before giving an explanation about why dogs failed on those tests.

It is not clear why the olfactory mirror test is controversial. This is a very relevant background for your study; it is necessary to discuss this in more depth.

L 94, it is hard for dogs to pay attention to the videos. Many dogs are discarded from the samples when the designs include images (photos or videos). Please, include this limitation regarding to your methodology.

It calls my attention that you chose the emotion of jealousy. This is an emotion little studied in dogs, the information about its physiological correlates is scarce and it does not imply any kind of self-recognition. Moreover, you said that “We focused on the jealousy paradigm, which we expect to drive autonomic nervous system activity in dogs” but there is no reference for this claim. Please justify the advantages of choosing this emotion. Finally, it is important to discuss that, considering that you did not find any indication of jealousy, the lack of this emotional component could affect the results. Meaning that if the situation was no relevant for the dogs, there was no motivation during the task.

“…dogs see their image in the mirror with motor-contingency as the self or another subject” What do you mean?

L 137 there are several studies that used HRV as an indicator of stress in dogs. Why did you mention only this one?

L 144 You said that it has been shown a decrease in SDNN during positive stimuli and a decrease in RMSSD during negative stimuli. However, in the discussion (L 394) you stated that decreased RMSSD and decreased SDNN are associated with stress states. It seems contradictory; please explain better the meaning of those parameters.

“This is because we believe that emotional changes, such as jealousy, are influenced by the temperament of the dog”. What do you mean? The word “believe” is confusing. Which is the evidence underpinning this statement? Or is it just a hypothesis? You can include more evidence about the relationship of temperament and social behavior in dogs using the CBARQ.

L 150 It is not clear here which dimensions of the CBARQ were analyzed.

L 157 “we examined whether the changes in HRV in dogs were due to their aggressive or timid temperament”. In my opinion your data can not allow you to establish a causal relationship between temperament and HRV.

L163 Was this sample the initial one or did you have to discard some dogs? Considering the dogs difficulty in watching videos it is unexpected that all the evaluated dogs completed the task.

L 178 You did not mention the presence of “non-owners” in the experiment during the introduction.

There is some important missing information. Specifically, how long had the dogs been living with their owners? How many dogs were living with other dogs in their household? This last factor must be included in the statistical analysis given that it could produce either, habituation or sensitization to the stimulus (owner interacting with another dog).

Please define better “familiar dog”. You mentioned that they shared time in the facility; did you mean the experimental facility? How long did the dogs interact with the other dog?

As far as I understand, you evaluated dogs in an unfamiliar location and the owners left the room during the procedure, leaving the dogs alone with an unknown person. This situation is stressful for the dogs (see all the results obtained in attachment tests). This could interfere with the reactions toward the videos. In addition, this could interfere with your resting assessment of the HRV. You have to discuss this important limitation.

Did you assess the time dogs spent watching each video? This could be a good indicator of the attention dogs paid to the test even in that stressful situation.

Did you counterbalance across dogs the order of presentation of the videos?

L 382 please include here the meaning of lower RMSSD SDNN

The conditions that you described in the method are: OW-A-INT, OW-S-INT, OW-A-IGN and NOW-A-INT. However, in L 396 you mention NOW-A-IGN, it was probably a typing error.

L 393-394 Please integrate this with the previous paragraph

You must include a discussion about why you found differences between NOW-A-INT and OW-S-INT but not between OW-A-INT and OW-S-INT

L 430 Include here the meaning of higher meanRRI

L415, It is not clear the definition of “status of the dog”

L 417 I would be enriching if you can include some behavioral assessment of fear.

L 431 Is there any other evidence of this statement in a situation in which there is no food? There are many other processes related to the food beyond its appetitive value.

L 455 “this experimental design can effectively examine whether dogs respond in a way that distinguishes between themselves and others”. You mentioned several limitations of this design. In addition, the unfamiliarity of the place and the person present during the test could have diminished its effectiveness. Therefore, I think that this method is potentially useful but it needs some improvements.

S1_Fig is not clear, please replace it

Reviewer #3: This is an interesting question. The introduction is quite detailed and well written. However, the authors have mostly compared human infants and dogs. I think some work on other primates should be cited here. There has been a lot of work carried out in this area in different primate species, like chimpanzees, bonobos, orangutans, etc. In fact, the mirror test is widely used across species, dolphins, elephants, pigeons, and many others have been tested. There should be some discussion about this in the introduction.

My major concern is the sample size. Only 12 dogs have been tested, and they are from different breeds.

The owner playing with other dog videos are my second point of concern, as the authors have stated that they could not control for breed and size. This might influence the focal dog’s response to the videos. Even if a dog is unable to identify itself in the video, it surely would be able to judge if the dog on the screen is completely different from itself!

The C-BARQ scores are used for analysis here, to correlate the temperament of the dogs with the HRV scores during the experiment. Similar analysis for the correlation, if any, of the scores with the baseline data should be carried out. The assumption here is that the owners’ perception of aggression/fear is accurate which is not necessarily correct, and this should be mentioned as a caveat.

As the authors themselves mention in the discussion, it is difficult to understand from this experiment, whether the dogs are responding to the humans or the dogs in the videos. This is a major drawback of the study. Moreover, it is highly possible that the dogs are responding to the videos simply because of the novelty of the set-up, or as a territorial response.

6. PLOS authors have the option to publish the peer review history of their article (what does this mean?). If published, this will include your full peer review and any attached files.

Reviewer #1: **Yes: **Andrea Sommese, Ph.D.

Reviewer #2: No

Reviewer #3: **Yes: **Anindita Bhadra

---

## [Author Response · Author response to Decision Letter 0]

11 May 2022

We appreciate the time and effort you and each of the reviewers have dedicated to providing insightful feedback on how to enhance our paper. Thus, it is with great pleasure that we resubmit our article for further consideration. We have incorporated changes in line with the detailed suggestions you have graciously provided. Especially for the additional statistical analysis. Since the descriptions were not clear enough, we have made corrections and added explanations.

Comments from the Editor;

Thank you for your valuable comments on our initial MS, and we tried to revise our MS accodring to your and other reviewers' comments.

1)Will you extend the literature also to quick examples of other species (see the comments of Reviewer 3)?

Re: As suggested, we added some sentences in the Introduction. “Because the presence of MSR has long been confirmed only in humans and some nonhuman primates (ex. [3-6]), the evolutionary development of MSR was thought to be limited to lineages such as humans and other non-human primates. Recently, however, its presence has been confirmed in other animal species, such as magpies, dolphins, elephants, and fish [7-10], and it has begun to be proposed that self-awareness may be an example of convergent evolution, as it occurs independently in different species. In order to explore the detailed mechanisms by which this self-awareness has followed an evolutionary path and has emerged, a comparative study in a larger number of animal species is necessary”.

2) I agree with Reviewers 1 and 3 that the number of dogs used is a rather low. It’s a good idea to make a power analysis first, as suggested by Reviewer 1, otherwise the non-significant results were not convincing to such a few subjects.

Re: We agree that the number of dogs (n=12) was not enough to fully examine our hypothesis, however even with such a small number, statistical significances were detected. We could not add more dogs due to the pandemic and we mentioned this issue in Discussion. We also added the power analysis as you suggested (Supplementary Table S3). “And most importantly, the sample size was relatively small due to the pandemic. As S3 Table indicates, the effect size of the results of this study was not conclusive, but very suggestive, and needs to be validated with a larger sample size”.

3) There are many parts of the Methods to be clarified. In what order did you play the videos on the tested dog? (On random, in a pre-defined order?) You should design the statistical model for repeated measures, anyway.

Re: we added this information in the Method, “the sequence of the presentation of four video was pseudorandom”. The statistical model included the sequence of the trial, as you suggested.

4) It’s fine you mentioned the citation for the C-BARQ assessment. However, will you briefly describe the basic principles of this assessment for those who are not familiar with that method? Besides, I think that would be good for the other tests, too. For further details of the Methods improvement, follow the suggestions provided first by Reviewer 2. But many comments by the other two reviewers are also useful.

Re: As you and other reviewers suggested, we added the information about C-barq and revised methods. “The various items and subscale scores of the C-BARQ have been shown to provide an accurate measure of the dog's behavioral phenotype.　The C-BARQ assessment yields multiple temperament factors in dogs. The study used scores for dog-directed aggression, dog-directed fear, and attachment or attention-seeking behavior. The reasons for using these indices are that we are interested in the possibility that the subject dog is reacting to other dogs with strong interest (aggression or fear), not just that the strength of the attachment may incite more jealousy”.

5) I have a problem myself with your statistics. You should first list all variables involved with some basic information, such as mean, SD, or SE, or min-max in countable variables and the levels of categorical variables. You haven’t mentioned potential effect of age, sex, and breed in your analysis (and how you coped with that). Before any other analyses, make the detection of multicollinearity. It’s most likely you did it so. However, when you mention multicollinearity as the last part of the Statistical analysis of the Methods, it is not clear how you did it. Still, I don’t think testing the multicollinearity by using the variance inflation factor only is sufficient. (See, for example, https://www.r-bloggers.com/2018/08/dealing-with-the-problem-of-multicollinearity-in-r/). For an extended period, statisticians have not recommended the stepwise method for studies like this (e.g., Derksen, S. & Keselman, H. J., 1992. Br. J. Math. Stat. Psychol. 45, 265-282; Whittingham, M. J., Stephens, P. A., Bradbury, R. B. & Freckleton, R. P., 2006. J. Anim. Ecol. 75, 1182-1189; etc.). Instead of using the stepwise procedure, you should first formulate the hypothesis (or hypotheses) according to which you should then construct the statistical model to be tested. (See for inspiration, for example Kleinbaum, D. G., Kupper, L. L., Muller, K. E. & Nizam, A., 2013. Applied regression analysis and other multivariable methods. Duxbury Press, Pacific Grove.)

Re: This is an important point. As you suggested we modified the tables to show basic values information. We changed the statistical method as you suggested. First we set our hypothesis in the Introduction, and excluded the stepwise procedure. We included the information of sex and age, but not breeds due to the number of dogs tested were not enough. We also checked the multicollinearity by using the variance inflation factor. “Emotional changes in the jealousy are influenced by the temperament of the dog [29,39-40]. The attention-seeking behaviors in this C-BARQ trait include pushiness and “jealousy” when attention is given to third parties. Therefore, it is highly likely that a dog's social responses will be varied by its individual temperament. In this study, we predicted individual differences in emotional responses, such as jealousy and disgust, to social interaction videos by dogs, and investigated the relationship between these responses and dogs' attachment to their owners using C-BARQ. In the same way, we examined whether the individual difference of the changes in HRV in dogs were correlated to their aggressive or fear temperament”. “In order to confirm the association between the C-BARQ temperament scores and the HRV data, a multivariate analysis was conducted using the NOW-A-INT, OW-A-INT, OW-S-INT, and the difference between the OW-S-INT and OW-A-INT values of HRV as dependent variables and the C-BARQ scores (dog-directed aggression, dog-directed fear, and attachment or attention-seeking behavior) were used as explanatory variable. For multivariate analysis, linear regression analysis was used. In order to remove the effect of multicollinearity from causing major problems in the results when performing linear regression analysis, the correlation between the explanatory variables was checked and the variance inflation factor was calculated and confirmed to be less than 10”.

Reviewer #1: 

The manuscript is interesting, and I liked the idea. I think it fits nicely on the topic of emotional response to social stimuli. Besides, it expands our knowledge on how dogs process and reacts to prerecorded videos.

There are a few things that are not quite clear to me but let's proceed with order.

INTRO:

The intro addresses the topic correctly. I believe that all the necessary literature has been cited and the initial hypothesis of the authors are clearly stated and easy to identify. Good idea to support behavioural data with a standardized questionnaire like the CBARQ.

Re: Thank you for your comment. Per another reviewer's comment, we have corrected the Intro.

METHODS:

Honestly, 12 is not a huge number and it would have been more interesting to have more subjects. I understand that sample size is an issue, especially during the pandemic but maybe I would have addressed this in the discussion. And/or you could add a power analysis showing that this sample is enough for the study.

Where the dogs intact or neutered? This factor should be also considered in the analysis.

Why did the authors not include also an unfamiliar dog? I would have definitely add more conditions (i.e., non-owner/owner interacting/non-interacting with unfamiliar dog).

Were non-owners friends of the owners, students, experimenters? It is not clear to me.

Re: We agree that the number of dogs (n=12) was not enough to fully examine our hypothesis, however even with such a small number, statistical significances were detected. We could not add more dogs due to the pandemic and we mentioned this issue in Discussion. We also added the power analysis as you suggested (Supplementary Table S3). “And most importantly, the sample size was relatively small due to the pandemic. As S3 Table indicates, the effect size of the results of this study was not conclusive, but very suggestive, and needs to be validated with a larger sample size”. In this study, under the context of jealousy, we expected emotional change caused by the interaction between the other dog and the owner. Therefore, we also added “We considered that a stranger dog would induce aggression and fear, so we used a familiar dog to elicit jealousy”. Of course, as you pointed out, we believe it is also necessary to add a stranger dog to the condition in future studies.

RESULTS:

It seems that here the results have been reported adequately.

Re: Thank you for your comment. We also added the results of statistical power analysis as mentioned above (Supplementary Table S3).

DISCUSSION:

I have to admit that this section was quite hard to follow.

As far as I understand the authors' findings are the following:

No differences at all in the mean RRI,

RMSSD different between non-owner interacting with another dog and owner ignoring the other dog,

RMSSD different between owner ignoring the other dog and interacting with their own dog,

SDNN same as RMSSD

I would maybe make a table with such results in a way that it makes it easier for the reader to interpret your findings. And/or I would divide the initial section of the discussion in paragraphs.

Re: As you suggested, we added the Supplementary table S3, in which statistical results were listed. We also added the first paragraph of Discussion describing the summary of the present results. Thank you.

Another thing that I think it is worth to mention in the discussion is that using questionnaires might be faulty to a degree. In previous studies employing the CBARQ the authors noticed how some people belonging to different cultures might have difficulties in expressing strong concepts like 'my dog NEVER/ALWAYS does X'. Hence, they tend to evaluate their pets in a more conservative way. I would add a line in the discussion addressing this.

Re: Thank you for pointing this out, and we added the following in the Discussion, “It should be however noted that the CBARQ may not have been assessed correctly by the owners”.

Reviewer #2: 

The Autonomic Nervous System Responses of Dogs to Human-Dog Interaction Videos manuscript studies the emotional response to social stimuli played on videos as well as the ability of dogs to distinguish their own image from that of other dogs. Both topics are interesting and they need further investigation. The methodology is original and highlights the relevance of using subtle physiological indicators of dogs’ emotions. However, there are some concerns that I detailed below.

Re: Thank you for your valuable comments, and we tried to revise our MS according to your comments.

Abstract: please include the definition of RRI

Re: corrected.

Introduction

Please clarify and further explain this statement “no successful cases of the mark test have been reported”. You should describe previous studies before giving an explanation about why dogs failed on those tests.

Re: We modified the sentence as you suggested, “However, no successful cases of the mark test have been reported [19]. Dogs may fail the mark test because of lack of motivation to investigate objects on their bodies”.

It is not clear why the olfactory mirror test is controversial. This is a very relevant background for your study; it is necessary to discuss this in more depth.

Re: Olfactory mirror test is a fascinating method to investigate the “self-odor recognition” in some species that do not rely on visual cues for individual recognition. While the mirror-self recognition test is said to determine that by touching it self's body through a mirror, one can determine that one recognizes oneself in the mirror. However, the olfactory mirror test is relatively difficult to determine whether one has just become habituated to one's own odor or whether it has recognized as a "part of it self". To avoid the misleading, we deleted this part of sentence.

L 94, it is hard for dogs to pay attention to the videos. Many dogs are discarded from the samples when the designs include images (photos or videos). Please, include this limitation regarding to your methodology.

Re: As you suggested, we added the words, “Although it is hard for dog to pay attention to images/videos”.

It calls my attention that you chose the emotion of jealousy. This is an emotion little studied in dogs, the information about its physiological correlates is scarce and it does not imply any kind of self-recognition. Moreover, you said that “We focused on the jealousy paradigm, which we expect to drive autonomic nervous system activity in dogs” but there is no reference for this claim. Please justify the advantages of choosing this emotion. Finally, it is important to discuss that, considering that you did not find any indication of jealousy, the lack of this emotional component could affect the results. Meaning that if the situation was no relevant for the dogs, there was no motivation during the task.

Re: As you suggested, we cited a reference in sentence, 29. Cook P, Prichard A, Spivak M, Berns GS. Jealousy in dogs? Evidence from brain imaging. Animal Sentience. 2018; 22(1). DOI: 10.51291/2377-7478.1319.

The advantage of the use of the jealousy paradigm is that self-recognition of dog’s own image can be verified by autonomic nervous system changes, by comparing of the autonomic changes induced by the other dog image.

As you pointed out, we could not find any indication of jealousy and the lack of this emotional component could affect the results. We think that this was because of the individual traits of dog, some dogs were more sensitive to the jealousy situation, but some were not. Therefore, we conducted the correlation analysis between attachment score and HRV parameters. As the results, we discovered that the difference of meanRRI between OW-S-INT and OW-A-INT condition correlated with the score of C-BARQ (attachment or attention-seeking behavior). This suggests that the more attached a dog is to its owner, the dog's emotional response to the difference between the self-video stimulus and the video stimulus of another dog was more distinct.

“…dogs see their image in the mirror with motor-contingency as the self or another subject” What do you mean?

Re: Sorry for this, and we revised the sentence as follows, “One question that arises is whether the responses to the video of itself would differ based on whether dogs see their image in the mirror as itself or another subject”.

L 137 there are several studies that used HRV as an indicator of stress in dogs. Why did you mention only this one?

Re: This reference was our previous study and we discovered that not only stress/negative emotion, but also positive emotion was able to be assessed by HRV.

L 144 You said that it has been shown a decrease in SDNN during positive stimuli and a decrease in RMSSD during negative stimuli. However, in the discussion (L 394) you stated that decreased RMSSD and decreased SDNN are associated with stress states. It seems contradictory; please explain better the meaning of those parameters.

Re: Two papers cited in the discussion reported the decrease of RMSSD and SDNN in the negative situation. However, the HRV can be easily modulated by the body movement and posture. In our previous study, we controlled the body movement and posture by selecting the time in which dogs were still and standing and revealed that only RMSSD decreased in the negative situation. 

“This is because that emotional changes, such as jealousy, are influenced by the temperament of the dog”. What do you mean? The word “believe” is confusing. Which is the evidence underpinning this statement? Or is it just a hypothesis? You can include more evidence about the relationship of temperament and social behavior in dogs using the CBARQ.

L 150 It is not clear here which dimensions of the CBARQ were analyzed.

Re: Thank you for pointing this out, and we rephased the sentences as follows, “Emotional changes in the jealousy are influenced by the temperament of the dog [29,39-40]. The attention-seeking behaviors in this C-BARQ trait include pushiness and “jealousy” when attention is given to third parties. Therefore, it is highly likely that a dog's social responses will be varied by its individual temperament. In this study, we predicted individual differences in emotional responses, such as jealousy and disgust, to social interaction videos by dogs, and investigated the relationship between these responses and dogs' attachment to their owners using C-BARQ.

L 157 “we examined whether the changes in HRV in dogs were due to their aggressive or timid temperament”. In my opinion your data can not allow you to establish a causal relationship between temperament and HRV.

Re: This is an important point and thank you for your suggestion. We revised the sentence as follows, “In the same way, we examined whether the individual difference of the changes in HRV in dogs were correlated to their aggressive or fear temperament.

L163 Was this sample the initial one or did you have to discard some dogs? Considering the dogs difficulty in watching videos it is unexpected that all the evaluated dogs completed the task.

Re: We added that 15 dogs were actually scheduled to participate. We also mentioned the dogs that were excluded.

L 178 You did not mention the presence of “non-owners” in the experiment during the introduction.

Re: ¬As suggested, we added the “non-owners” in the Introduction as follows, “As a control, the dog will not show such an emotional response when watching “non-owners” interacting with the “other dog”.

There is some important missing information. Specifically, how long had the dogs been living with their owners? How many dogs were living with other dogs in their household? This last factor must be included in the statistical analysis given that it could produce either, habituation or sensitization to the stimulus (owner interacting with another dog).

Re: We did not obtain these data and therefore could not statistically process them. We apologize for not being able to answer your concern.

Please define better “familiar dog”. You mentioned that they shared time in the facility; did you mean the experimental facility? How long did the dogs interact with the other dog?

Re: We are sorry for misleading. We revised the sentence as follows, “A familiar one was a dog that has spent time sharing the same environment (University or home), showing affiliative behavior such as greeting to the person”.

As far as I understand, you evaluated dogs in an unfamiliar location and the owners left the room during the procedure, leaving the dogs alone with an unknown person. This situation is stressful for the dogs (see all the results obtained in attachment tests). This could interfere with the reactions toward the videos. In addition, this could interfere with your resting assessment of the HRV. You have to discuss this important limitation.

Re: This is an important point. We excluded the dog from the experiment who we could not measure ECG accurately due to not showing resting behavior, such as standing and lying and after the owner left the dog. Even so, we cannot eliminate the possibility that subject dogs were under stressful condition during the resting period. We added the following in the Discussion, “HRV parameters in the resting time could not exactly reflect neutral emotional state due to being left in an unfamiliar environment with the experimenter”.

Did you assess the time dogs spent watching each video? This could be a good indicator of the attention dogs paid to the test even in that stressful situation.

Re: Thank you for your suggestion, We recorded the dog’s behavior but it was difficult to determine the duration of watching video. It was because the screen was relative wide and the dog’s visual field was varied by each dog (some had wide view and some had narrow view, depending on the shape of the head). This would be in future experiment. 

Did you counterbalance across dogs the order of presentation of the videos?

Re: Yes, and we added this information in the Method, “the sequence of the presentation of four video was pseudorandom”.

L 382 please include here the meaning of lower RMSSD SDNN

Re: As suggested, we changed the sentence as follows, “Specifically, RMSSD and SDNN were lower in NOW-A-INT and OW-S-INT than in OW-A-IGN (Fig 3). This is may be a result of higher arousal of dogs in the human-dog interacting stimuli”.

The conditions that you described in the method are: OW-A-INT, OW-S-INT, OW-A-IGN and NOW-A-INT. However, in L 396 you mention NOW-A-IGN, it was probably a typing error.

Re: Sorry for our mistake and we fixed it.

L 393-394 Please integrate this with the previous paragraph

Re: Thank you for your suggestion and we revised it.

You must include a discussion about why you found differences between NOW-A-INT and OW-S-INT but not between OW-A-INT and OW-S-INT

Re: We are sorry that we do not understand the point of your concern, Regarding HRV comparison, we did not see differences between NOW-A-INT and OW-S-INT. If you mentioned the sentences from L393 to L406 (previous version), we found that NOW-A-INT and OW-S-INT, but not OW-A-INT, was different from the control condition of OW-A-IGN. At this moment, we do not have clear explanation for this, and we added the following sentences in the Discussion, “it was difficult to explain why there was no difference between OW-A-INT and OW-A-IGN; OW-A-INT video also displayed a human interacting with the dog. One way to solve this discrepancy is analyzing which part of the video the dog was paying attention to. Particularly in the 3 interaction conditions, dog were more attentive to the video stimuli as compared to the OW-A-IGN. However, because of the technical aspects of this study, we were unable to do so”.

L 430 Include here the meaning of higher meanRRI

Re: We added the words “higher emotional arousal including positive state” in the sentence.

L415, It is not clear the definition of “status of the dog”

Re: We replaced the words with “visual information obtained from the dog in the video”.

L 417 I would be enriching if you can include some behavioral assessment of fear.

Re: Thank you for your suggestion, and we rephased the sentence as follows, “As described above, the dog subjects showed vigilant behavior, such as paying attention to the video toward the dog in the OW-S-INT condition”.

L 431 Is there any other evidence of this statement in a situation in which there is no food? There are many other processes related to the food beyond its appetitive value.

Re: We added our paper describing that dog showed increased heart rate, decreased parasympathetic activity during the reunion with the owner after a short time of separation (Nagasawa, M., Mogi, K., & Kikusui, T. (2009). Attachment between humans and dogs. Japanese Psychological Research, 51(3), 209-221).

L 455 “this experimental design can effectively examine whether dogs respond in a way that distinguishes between themselves and others”. You mentioned several limitations of this design. In addition, the unfamiliarity of the place and the person present during the test could have diminished its effectiveness. Therefore, I think that this method is potentially useful but it needs some improvements.

Re: Thank you for your pointing this out, and we rephased the sentence as follows, “this experimental design can potentially examine whether dogs respond in a way that distinguishes between themselves and others”. We also added the following sentence in the Discussion, “HRV parameters in the resting time could not exactly reflect neutral emotional state due to being left in the unfamiliar environment with the experimenter”.

S1_Fig is not clear, please replace it

Re: We replaced the Figure S1.

Reviewer #3: 

This is an interesting question. The introduction is quite detailed and well written. However, the authors have mostly compared human infants and dogs. I think some work on other primates should be cited here. There has been a lot of work carried out in this area in different primate species, like chimpanzees, bonobos, orangutans, etc. In fact, the mirror test is widely used across species, dolphins, elephants, pigeons, and many others have been tested. There should be some discussion about this in the introduction.

Re: Thank you for your valuable comments, and we tried to revise our MS according to your comments. We added reference regarding to mirror-self recognition test in some species in the Introduction, “Because the presence of MSR has long been confirmed only in humans and some nonhuman primates (ex. [3-6]), the evolutionary development of MSR was thought to be limited to lineages such as humans and other non-human primates. Recently, however, its presence has been confirmed in other animal species, such as magpies, dolphins, elephants, and fish [7-10], and it has begun to be proposed that self-awareness may be an example of convergent evolution, as it occurs independently in different species. In order to explore the detailed mechanisms by which this self-awareness has followed an evolutionary path and has emerged, a comparative study in a larger number of animal species is necessary.

My major concern is the sample size. Only 12 dogs have been tested, and they are from different breeds.

Re: As the other reviewers pointed out, our experiment limitation was a small sample size. We agree that the number of dogs (n=12) was not enough to fully examine our hypothesis, however even with such a small number, statistical significances were detected. We could not add other dogs due to the pandemic and we mentioned this issue in Discussion. We also added the power analysis as you suggested (Supplementary Table S3). “And most importantly, the sample size was relatively small due to the pandemic. As S3 Table indicates, the effect size of the results of this study was not conclusive, but very suggestive, and needs to be validated with a larger sample size”

The owner playing with other dog videos are my second point of concern, as the authors have stated that they could not control for breed and size. This might influence the focal dog’s response to the videos. Even if a dog is unable to identify itself in the video, it surely would be able to judge if the dog on the screen is completely different from itself!

Re: This is an important issue and thank you for pointing this out. To solve this concern, additional experiment should be conducted to determine which parts of the video the dogs were paying attention to and how their HRV changed during the video. We described the followings in Discussion, “But in this case, it was difficult to explain why there was no difference between OW-A-INT and OW-A-IGN; OW-A-INT video also displayed human interaction with the dog. One way to solve this discrepancy was analyzing which part of the video the dog was paying attention to. Particularly in the 3 interaction conditions, dogs were more attentive to the video stimuli as compared to the OW-A-IGN. However, because of the technical aspects of this study, we were unable to do so. Therefore, it is difficult to conclude that the dog's emotional change was caused by seeing the state of another dog, rather than the state of the human”.

The C-BARQ scores are used for analysis here, to correlate the temperament of the dogs with the HRV scores during the experiment. Similar analysis for the correlation, if any, of the scores with the baseline data should be carried out. The assumption here is that the owners’ perception of aggression/fear is accurate which is not necessarily correct, and this should be mentioned as a caveat.

Re: As you suggested we conducted correlation analysis between HRV in the basal condition and C-barq scores, but there was no significance. As you mentioned, we added the sentence in the Discussion, “It should be noted, however, that the CBARQ may not have been assessed correctly by the owners”.

As the authors themselves mention in the discussion, it is difficult to understand from this experiment, whether the dogs are responding to the humans or the dogs in the videos. This is a major drawback of the study. Moreover, it is highly possible that the dogs are responding to the videos simply because of the novelty of the set-up, or as a territorial response.

Re: Similar concern was raised by the reviewer #2. We recorded the dog’s behavior but it was difficult to determine the duration of the dog watching the video. It was because the screen was relatively wide and the dog’s visual field was varied by each dog (some had wide view and some had narrow view, depending on the shape of the head). This would be in future experiment. We also included the following sentence in the Discussion, “and HRV parameters in the resting time could not exactly reflect neutral emotional state due to being left in an unfamiliar environment with the experimenter”.

---

## [Decision Letter · Decision Letter 1]

22 Sep 2022

PONE-D-21-29226R1Autonomic Nervous System Responses of Dogs to Human-Dog Interaction VideosPLOS ONE

Dear Dr. Kikusui,

Thank you for submitting your manuscript to PLOS ONE. After careful consideration, we feel that it has merit but does not fully meet PLOS ONE’s publication criteria as it currently stands. Therefore, we invite you to submit a revised version of the manuscript that addresses the points raised during the review process.

Based on my own reading, the manuscript seems almost ready. I have no concerns at al, but I’d highly suggest the authors to report all stats parameters (CIs, effect sizes, assumptions check etc.) 

We look forward to receiving your revised manuscript.

Kind regards,

Thiago Fernandes, MS, EbS, Sp. Neur, PhD

Academic Editor

PLOS ONE

Journal Requirements:

Reviewers' comments:

Reviewer's Responses to Questions

**Comments to the Author**

1. If the authors have adequately addressed your comments raised in a previous round of review and you feel that this manuscript is now acceptable for publication, you may indicate that here to bypass the “Comments to the Author” section, enter your conflict of interest statement in the “Confidential to Editor” section, and submit your "Accept" recommendation.

Reviewer #1: All comments have been addressed

Reviewer #2: All comments have been addressed

2. Is the manuscript technically sound, and do the data support the conclusions?

Reviewer #1: Yes

Reviewer #2: (No Response)

3. Has the statistical analysis been performed appropriately and rigorously? 

Reviewer #1: I Don't Know

Reviewer #2: (No Response)

4. Have the authors made all data underlying the findings in their manuscript fully available?

Reviewer #1: Yes

Reviewer #2: (No Response)

5. Is the manuscript presented in an intelligible fashion and written in standard English?

Reviewer #1: Yes

Reviewer #2: (No Response)

6. Review Comments to the Author

Reviewer #1: Dear authors,

thanks for addressing my comments and those of the other reviewers (and the editor).

I am satisfied with the replies and modifications you provided. One suggestion, it would be easier if next time you'll indicate the lines where you modified your manuscript in our response.

Reviewer #2: The authors have made most of the requested modifications. In my opinion the manuscript is suitable for publication.

7. PLOS authors have the option to publish the peer review history of their article (what does this mean?). If published, this will include your full peer review and any attached files.

Reviewer #1: **Yes: **Andrea Sommese, Ph.D.

Reviewer #2: No

---

## [Author Response · Author response to Decision Letter 1]

11 Oct 2022

We appreciate the time and effort you and each of the reviewers have dedicated to providing insightful feedback on how to enhance our paper. We have noted below the points you have raised.

Reviewer #1: Dear authors,

thanks for addressing my comments and those of the other reviewers (and the editor).

I am satisfied with the replies and modifications you provided. One suggestion, it would be easier if next time you'll indicate the lines where you modified your manuscript in our response.

Re: Thank you for your comment. The lines of the manuscript modified are listed below.

The modified lines in Abstract are 42-46,48.42-46,48.

The modified lines in Introduction are 79-86,95,96,98,113,134,135,145,146,163-177.

The modified lines in Method are 186-188,234,235,248-252,311-315,317,318,352-358.

The modified lines in Results are 379-386,388,389 and Table2, Table3, S3 Table, S4 Table.

The modified lines in Discussion are 411-413,419,420,422,425-435,439-444,450-453,455,456,468,470-472,491-496.

Also, the references added are 4-10,39,51. The references have been renumbered accordingly.

---

## [Editor Report · Decision Letter 2]

13 Oct 2022

PONE-D-21-29226R2Autonomic Nervous System Responses of Dogs to Human-Dog Interaction VideosPLOS ONE

Dear Dr. Kikusui,

Thank you for submitting your manuscript to PLOS ONE. After careful consideration, we feel that it has merit but does not fully meet PLOS ONE’s publication criteria as it currently stands. Therefore, we invite you to submit a revised version of the manuscript that addresses the points raised during the review process.

ACADEMIC EDITOR: Please check my comments below, since they were overlooked in your edits.

A rebuttal letter that responds to each point raised by the academic editor and reviewer(s). You should upload this letter as a separate file labeled 'Response to Reviewers'.A marked-up copy of your manuscript that highlights changes made to the original version. You should upload this as a separate file labeled 'Revised Manuscript with Track Changes'.An unmarked version of your revised paper without tracked changes. You should upload this as a separate file labeled 'Manuscript'

We look forward to receiving your revised manuscript.

Kind regards,

Thiago P. Fernandes 

Academic Editor

PLOS ONE

Journal Requirements:

Additional Editor Comments (if provided):

Dear authors,

I think my comments were overlooked and, hence, I am sending them again.

They are easy-to-solve and very simple. Previous comments:

Based on my own reading, the manuscript seems almost ready. I have no concerns at al, but I’d highly suggest the authors to report all stats parameters (CIs, effect sizes, assumptions check etc.)
---

## [Author Response · Author response to Decision Letter 2]

17 Oct 2022

We appreciate the time and effort you and each of the reviewers have dedicated to providing insightful feedback on how to enhance our paper. We have reviewed the reference lists and ensured that it is complete and correct. The references added are 4-9,38,49, and the reference deleted is the previous number 32 because of changes in the text. The references have been renumbered accordingly. We also added the normality test statistic to line 327 and the effect sizes and Cl to lines 355-359 (also S3Table) of the revised manuscript. Other points raised are noted below.

From Editor and Editorial office.

Re: Thank you for your comment. As pointed out by the reviewer, added a reference on self-recognition in other animals (References list: 4-9) to the Introduction. Also, as pointed out by the reviewer, we added evidence on the relationship between temperament and social behavior in dogs (References list: 38) and changed the text accordingly, so we deleted reference list-32 from the Introduction. As well, the reviewer pointed out, we added a reference that the dog showed increased heart rate, and decreased parasympathetic activity during the reunion with the owner after a short time of separation (References list: 49) to the Discussion.

I’d highly suggest the authors to report all stats parameters (CIs, effect sizes, assumptions check etc.)

Re: Thank you for your comment. We included the statistical parameters as you suggested, and we revised the tables.

From Reviewer #1: 

Thanks for addressing my comments and those of the other reviewers (and the editor).

I am satisfied with the replies and modifications you provided. One suggestion, it would be easier if next time you'll indicate the lines where you modified your manuscript in our response.

Re: Thank you for your comment. The lines of the manuscript modified are listed below.

The modified lines in Abstract are 42-46,48.

The modified lines in Introduction are 79-86,95,96,98,113,134,135,145,146,163-177.

The modified lines in Method are 186-188,234,235,248-252,311-315,317,318,352-358,340.

The modified lines in Results are 371,373,379-386,388,389 and Table2, Table3, S3 Table, S4 Table.

The modified lines in Discussion are 411-413,419,420,422,425-435,439-444,450-453,455,456,468,470-472,491-496.

---

## [Editor Report · Decision Letter 3]

19 Oct 2022

Autonomic Nervous System Responses of Dogs to Human-Dog Interaction Videos

PONE-D-21-29226R3

Dear Dr. Kikusui,

We’re pleased to inform you that your manuscript has been judged scientifically suitable for publication and will be formally accepted for publication once it meets all outstanding technical requirements.

Kind regards,

Thiago P. Fernandes, PhD

Academic Editor

PLOS ONE

Additional Editor Comments (optional):

Thank you for your thoughtful edits.
---

## [Editor Report · Acceptance letter]

26 Oct 2022

PONE-D-21-29226R3 

Autonomic Nervous System Responses of Dogs to Human-Dog Interaction Videos 

Dear Dr. Kikusui:

I'm pleased to inform you that your manuscript has been deemed suitable for publication in PLOS ONE. Congratulations! Your manuscript is now with our production department. 

Kind regards, 

on behalf of

Dr. Thiago P. Fernandes 

Academic Editor

PLOS ONE